# Estimating a panel MSK dataset for comparative analyses of national absorptive capacity systems, economic growth, and development in low and middle income countries

**Muhammad Salar Khan** *

Schar School of Policy and Government, George Mason University, Arlington, Virginia, United States of America

* mkhan63@gmu.edu

## Abstract

Within the national innovation system literature, the low- and middle-income countries (LMICs) eligible for the World Bank's International Development Association (IDA) support, are rarely part of empirical discourses on growth, development, and innovation. One major issue hindering empirical analyses in LMICs is the lack of complete data availability. This work offers a new full panel dataset with no missing values for IDA-eligible LMICs. I use a standard, widely respected multiple imputation method (specifically, *Predictive Mean Matching*) developed by Rubin in the 1980s, which conforms to the structure of multivariate continuous panel data at the country level. The incomplete input data consisting of many variables come from publicly available established sources. These variables, in turn, capture six crucial country-level capacities: technological capacity, financial capacity, human capital capacity, infrastructural capacity, public policy capacity, and social capacity. Such capacities are part and parcel of the *National Absorptive Capacity Systems* (NACS). The dataset (MSK dataset) thus produced contains data on 47 variables for 82 LMICs between 2005 and 2019. The dataset has passed a quality and reliability check and can therefore be used for comparative analyses of national absorptive capacities and development, transition, and convergence among LMICs.

## 1. Introduction

*Without data, you're just another person with an opinion.*

(*William Edwards Deming*)

The National Innovation System (NIS) focuses on a broad range of variables, activities, institutions, and their interactions that can foster economic growth and development in countries

**Data Availability Statement:** All relevant data are within the paper and its Supporting information files.

**Funding:** The author(s) received no specific funding for this work.

**Competing interests:** The author has declared that no competing interests exist.

[1]. However, this literature underrepresents the global South. One of the major problems for this lack of reasonable representation stems from the lack of data for low- and middle-income countries (LMICs). By resulting in the exclusion of LMICs in empirical analyses, missing data lead to either positively or negatively biased results that manifest themselves in over and underestimated effect sizes.

Despite the general limitations, several studies have recently investigated NIS and its relationship with growth and development in some developing economies [2–5]. Other studies, using capacities as a way to operationalize NIS, have employed available data for diverse samples of countries to estimate the quantitative impact of financial, technological, and social capacities of countries on their economic growth and development process [6–11].

Inspired by the studies on capacities and economic development, Khan [12] has recently rigorously operationalized a thorough list of capacities that capture innovation, knowledge absorption, and learning processes in LMICs and further included those capacities in a formal framework of National Absorptive Capacity System (NACS). A firm-level concept of "absorptive capacity," as advanced by Cohen and Levinthal [13], particularly motivates the NACS framework. As a modified version of NIS, NACS considers an LMIC an "economic learning" entity that absorbs, creates, and deploys knowledge, learning, and skills subject to the strength of its local capacities [14].

To study NACS and its evolution in LMICs and to further examine the impact of the framework capacities on economic development in LMICs, complete panel data (country-year observations) on variables that measure capacities are required. Unfortunately, such variables are not wholly available across LMICs eligible for the World Bank's International Development Association (IDA) support. The IDA eligibility depends mainly on a country's relative poverty, defined as the Gross National Income (GNI) per capita below an established threshold updated annually (1,185 US dollars in the fiscal year 2021). IDA also supports some countries, including several small island economies, that are above the operational cutoff but lack the creditworthiness needed to borrow from the International Bank for Reconstruction and Development (IBRD). Since IDA eligibility is based on GNI per capita, countries graduate and reinter (More information on IDAs can be accessed here: https://ida.worldbank.org/en/about/borrowing-countries). Such IDAs are the foci of this study (I have data on 82 countries—74 among them are still eligible for IDA resources and 8 countries recently graduated, together I call them LMICs). Since the LMICs are data-impoverished, there is a dire need to fix the problem of missing data for those LMICs, presumably prime candidates for development, learning, and innovation. Therefore, in this article, I build a complete and recent dataset on variables constituting capacities within LMICs, using established statistical and machine learning techniques.

Data incompleteness, commonly called the missing data problem, severely hampers empirical research. Various research fields have extensively investigated missing data dynamics, consequences, and possible remedies [15–20]. However, the innovation system and absorptive capacity literature have yet to thoroughly investigate missing data's nuances, processes, and implications. One significant repercussion of missing data is that the current empirical literature on NIS and economic growth suffers from an imbalance. The literature either focuses on many countries within a limited period [7] or analyzes a few economies for an extended time [21, 22]. The former strand of literature can only provide a limited study of the evolution within NIS and NACS, whereas the latter prevents analyses in many LMICs. Hence neither is ideal; while the former is static, the latter is not representative of the LMICs.

This article systematically compiles, estimates, and imputes an incomplete dataset to alleviate the missing data problem in LMICs eligible for IDA support. It employs multiple imputation (MI) approach that *efficiently* and *consistently* estimates missing data and generates a

panel dataset for 82 LMICs between 2005 and 2019. MI uses state-of-the-art statistical methods to address the missing data problem [18, 23]. By treating missing variables as outcomes and complete variables as predictors, MI statistical methods either impute all incomplete variables in a single computation step (multivariate regression model) or impute one variable at a time in a series (univariate regression models). Many research fields in physical and biological sciences have embraced such techniques [24–27]. This work explicitly employs univariate regression modeling, a variable-by-variable (sequential or chained) predictive mean matching (PMM) technique [28]. As an MI conditional modeling approach, PMM imputes missingness dependent on observed data in continuous, panel variables that do not have to be normally distributed [28–30]. This technique returns meaningful imputations that respect the data distribution of the original incomplete dataset (observed dataset).

Castellacci and Natera [31] conducted a similar data compilation study (CANA hereon). The researchers estimate a CANA dataset for 134 countries between 1980 and 2008 using an MI algorithm developed by Honaker and King [32]. The proposed MSK dataset is similar to CANA dataset as both are panel datasets estimated using novel MI techniques. Similarly, both datasets have a roughly identical structural build of NACS and NIS. For instance, they contend that such systems are measured by dimensions (CANA) and capacities (MSK), which, in turn, are captured by many variables interacting in multiple ways.

Although this article builds on CANA, it is different in several ways. First, as opposed to the CANA dataset, the MSK dataset estimated here focuses on relatively more data-deficient and economically poor IDA-eligible countries.

Secondly, though the MSK dataset employs some CANA dataset variables, it has an entirely different functional and operational conception of the capacities and the variables used to operationalize those capacities. Particularly, Public Policy and Social Capacity are operationalized very differently. In the MSK dataset, the Public Policy Capacity includes variables about public sector management and institutions, economic management, structural policies, the strength of legal rights, and statistical capacity scores of countries, whereas the Social Capacity includes variables on policies for social inclusion, human resource rating, social protection rating, equity of public resource use, poverty headcount ratio, and social contributions. On the other hand, in the CANA dataset, the analogous dimensions are the Political-institutional dimension (which comprises freedom of press and speech, human rights, women's rights, and political rights, among other factors) and the Social capital dimension (which includes the importance of friends, family, marriage, trust, happiness, and Gini Index).

Additionally, the MSK dataset includes an extended set of other relevant variables to measure capacities. The MSK dataset consists of 47 variables for all economies in the dataset. In contrast, CANA consists of 34 variables for all economies and another seven variables for a restricted set of countries within the dataset.

Fourth, the timeframe for this study is truncated to fifteen years, not only because it is a decent period for panel analysis but also because of pragmatic concerns regarding data availability, particularly on public and social policy capacity variables. The World Bank Group's country offices started collecting these variables in the IDA-eligible countries from 2005 onwards [33].

The last vital distinction worth considering is that the CANA dataset is estimated using Honaker and King's [32] Expectation-Maximization algorithm. The MSK, on the other hand, is estimated using the Multiple Imputation by Chained Equations Predictive Mean Matching (*MICE PMM*) algorithm. Although the *EM* algorithm is efficient and undoubtedly suitable for panel data, it forces a normal distribution on the imputed data regardless of the distribution structure (skewed, unimodal, bimodal) in the observed data [34]. In contrast, the MICE PMM algorithm preserves the distribution pattern of observed data in the imputed values [35], and it

has been used for panel data imputation [36]. Besides preserving the distribution pattern in the imputed values, the MICE PMM is best suited for this study because the data structure is heteroskedastic (Variances of the variables in data mostly differ; for instance, variance for *days to enforce a contract* is 80 times larger than the variance for *days to start a business*) and associations among variables are nonlinear as can be seen in scatterplots.

In short, this article contributes to the literature by constructing a complete dataset and establishing its relevance for panel analyses of NACS and economic growth, among other analyses, in LMICs. A standard MICE PMM algorithm is employed to construct this dataset. The panel dataset, hence obtained, is complete with no missing values. It consists of 47 variables grouped into six vital capacities for each country: technological capacity, financial capacity, human capital capacity, infrastructural capacity, public policy capacity, and social capacity. The incomplete (original or observed) dataset, which contains many missing values, is constructed from reputable data sources such as the World Bank, International Monetary Fund (IMF), International Labor Organization (ILO), United Nations COMTRADE, and United Nations Educational, Scientific and Cultural Organization (UNESCO), among others (see S2 Table). The MSK dataset is estimated from this observed dataset, which provides information on 82 LMICs between 2005 and 2019 (total observations are 1,230 country-year observations). A four-way quality check establishes this dataset's reliability and usefulness for researchers interested in panel analyses of absorptive capacity and innovation system, economic development, economic policy, and convergence analysis within LMICs.

The rest of the paper is shaped as follows. Section 2 gives a brief literature landscape, the association between NIS and NACS, and discusses the missing data and its implications on methodologies. Section 3 further discusses the importance of handling missing data, strategies to address missingness, and underlying missing data mechanisms. Section 4 elaborates on Multiple Imputation and MICE PMM technique. Section 5 discusses the MSK dataset and the steps taken to develop this dataset. Section 6 carries out a brief descriptive analysis of the MSK dataset, and Section 7 conducts a quality check of the estimated dataset. Lastly, Section 8 concludes by summarizing the results and implications of this work. The Supporting Information includes graphs and tables, conveying more information on how the database is constructed and other dataset characteristics.

## 2. From NIS to NACS: Comparative analyses of national systems and growth, and development and the problem of missing data in LMICs

The concept of NIS emerged in the 1990s [37–39]. It considers systems, activities, institutions, and interactions as the driving force behind economic growth and development [1, 40]. The strength of these factors explains cross-country differences in growth, development, and innovation. Around the time NIS emerged, Cohen and Levinthal developed the idea of "absorptive capacity" to explain how learning is consolidated in a firm and how it impacts its growth [13]. In the early 2000s, researchers extended the firm-level concept to a national level [41, 42]. They developed a theoretical framework for aggregating national absorptive capacities upwards from a firm level. Other empirical studies also applied the idea nationally [8]. These works used different capacities emerging in NIS literature (such as technological and social capacities) as proxies for national absorptive capacity. In this essence, NACS is essentially an offshoot of NIS.

Earlier, foundational theoretical and empirical work on NIS focused mainly on prosperous economies [37, 43]. Later, NIS literature theoretically included developing countries, as they considered developing countries "national economic learning" entities and "imitation" centers

[4, 44, 45]. National-level capacities literature examining the impact of capacities on economic development also included some developing economies in their analyses [8]. However, because of the lack of data in LMICs, such studies had to compromise operationalizing the complex and multifaceted capacities proposed in NIS and NACS. Similarly, the lack of data on many vital variables perhaps trimmed the list of essential capacities in their analyses.

Another critical challenge that missing data poses is limiting the application of study methodologies in many LMICs. In general, quantitative studies of capacities and development use mainly two different methodologies: panel regression analyses and composite indicator analyses.

Panel regression analyses examine the empirical relationship between a few capacity variables and comparative national differences in GDP per capita growth across countries [46, 47]. While powerful as they consider the dynamic nature of capacities, such panel studies ignore or drop off many LMICs because longitudinal data for many variables are missing in these countries. As a result, the coefficients of interest obtained through panel analyses do not provide information about the economically poor economies. Using econometric terminology, the estimates from such studies exhibit an upward or downward bias by overestimating or underestimating the effect of *capacities* on *economic growth*.

On the other hand, composite indicator analyses establish a country's comparative standing against other countries by building aggregate or composite indicators that denote different dimensions of technological and social capabilities [7, 48]. Compared to panel analyses, the composite analyses consider many countries, including some LMICs. However, since most LMICs have limited data, such studies are usually static (one-year studies), ignoring how NACS evolved. Also, not all LMICs have data on all the variables of interest available for one particular year. Therefore, even composite analyses cannot possibly include all LMICs.

Generally, data availability restricts the number of countries and periods used in the analyses. Both methodologies are challenging for developing countries, particularly LMICs eligible for IDA resources, which are the foci of this study. This article contributes to alleviating the problems stemming from missingness by constructing a new complete panel dataset. A statistical technique called MICE PMM is employed to estimate the missing values in the original incomplete data sources [23]. Out of many imputation suites, this article considers MICE PMM because they are powerful, efficient, consistent, convenient, and reliable. The following section elaborates on why it is essential to adequately handle missing data and what strategies could be used to deal with missing data.

## 3. Properly handling missing data- why it is crucial, mechanisms underlying missing data, and strategies to handle missing data

It is essential to carefully consider the missing data problem to obtain accurate estimates of the parameters of interest in any analysis. Missing data pose many dilemmas in data analysis. The chief dilemma is that if a researcher uses original data by excluding subjects with missing data from the study, the researcher will not use all the existing information in the data, most likely causing over- or underestimated parameters (aka 'biased parameters'). To treat bias in parameters due to the exclusion of subjects in the analysis, a researcher can impute the missing data. During the imputation process, however, the researcher should take utmost care in preserving variability found in existing data and incorporating uncertainty underlying any missing data. Therefore, employing proper and standard imputation methodologies is imperative to estimate a reliable dataset.

Provided that the imputation technique is sound, one may get reliable imputations. The first step in getting the imputation technique right essentially means being mindful of the

missing data pattern and what might have caused it. The literature considers three potential mechanisms underlying missing data [49].

### Missing Completely At Random (MCAR)

Missing is MCAR if it is genuinely by chance, i.e., missingness is independent of data characteristics. In other words, missingness in MCAR is not related to any nonmissing or missing values in the data set. For example, the random loss of a blood sample in the lab suggests MCAR.

### Missing At Random (MAR)

Data exhibits MAR if the missingness is due to observed but not unobserved data. In other words, the observed data explains the missingness. For example, women may be less likely to report their age, regardless of their actual age.

### Missing Not At Random (MNAR)

In such a mechanism, missing values explain missingness. For example, individuals with higher salaries may be less willing to answer survey questions about their pay. Another example of MNAR relates to a person not attending a drug test because they took drugs the night before.

Understanding the mechanisms underlying missing data is extremely important to properly handle data. If a researcher fails to understand the missing data pattern and the underlying mechanism and imputes missing values, the missing data may be mistreated. Consequently, results will exhibit insufficient statistical power, upward or downward biases in parameters of interest, under or overestimated standard errors of the parameters, and other inaccurate findings.

Two main strategies are employed to handle missing data: 1) deletion and 2) substitution and imputation [50]. Deletion (also called complete or available-case analysis) is of two kinds: *pairwise* or *listwise* deletion [51]. Both these kinds exclude observations with missing values while analyzing data [51]. Imputation or substitution imputes or substitutes for missing values, and it is also of two main types: single imputation and multiple imputation [19].

Single imputation produces one complete dataset when imputing for missing values. It can be accomplished via several techniques such as mean substitution, mode substitution, nearest neighbor-based imputation, regression, or cold deck imputation [52]. Multiple Imputation (MI), on the other hand, produces multiple imputed data sets, employs a statistical analysis model to each one, and eventually merges all analysis results to generate an overall result [18].

Based on various data pattern assumptions and underlying data structures, MI is executed in many ways, such as parametric approaches (*Multivariate Normal MI*) that work well with normally distributed data or semiparametric approaches (*Multiple Imputation by Chained Equations* including *Predictive Mean Matching*) that relax normality assumption (Please see Pace [53] for more details on parametric vs. semiparametric approaches). Another imputation technique, performed in one or many runs, is Expectation-Maximization (EM) algorithm. EM is an iterative algorithm that finds maximum likelihood estimates in parametric models [54]. These strategies have both pros and cons (see S1 Table). Of those strategies, this article employs Multiple Imputation by Chained Equations (MICE), specifically Predictive Mean Matching (PMM), for imputing missing values that do not observe a normal distribution. MICE PMM is not only a convenient, standard, and reliable technique but also gives very accurate and plausible estimates for the data under consideration [36, 55]. The next section briefly describes MI, MICE, and PMM.

## 4. The multiple imputation method and predictive mean matching

Rubin [56] first introduced the multiple imputation methodology as an *efficient* statistical methodology to estimate missing values in a dataset. Several other researchers also explain this technique [56, 57]. Over the years, this methodology has evolved into various methods, catering to missingness in diverse data models. MI overcomes many of the problems associated with deletion and other single imputation techniques [58, 59]. In addition, the methodology returns efficient and accurate estimates and preserves *variability*, which is otherwise lost using other single imputation techniques (such as mean or cold deck imputation).

MI is valid under MAR (*Missing at Random*) assumption [59]. Therefore, MI estimates missing values using available, observed data [60].

Since there is uncertainty about missing data values, the estimation process is repeated *m* times (this step refers to the *imputation stage*). From the imputation stage, *m* complete datasets are generated. In the next stage (*analysis stage*), econometric analyses of interest are separately performed on *m* datasets. Finally, all these multiple results are combined (pooled) to obtain a final value of the coefficient of interest, for instance, regression coefficients (*pooling stage*). In short, a standard MI process produces multiple imputed datasets, applies a statistical analysis model to each dataset, and then integrates all analysis results to create an overall result (see Fig 1 below).

Suppose the imputation model at the imputation stage is specified correctly and the data exhibit a normal distribution. In that case, MI yields consistent parameter estimation and confidence intervals that incorporate uncertainty because of the missing data [29]. To clarify, the correct specification of an imputation model entails the inclusion of variables considered to predict missingness and variables associated with the variable being imputed, and the outcome variable of the analysis model [29, 61].

One of the common parametric approaches for MI execution is Multivariate Normal distribution MI (*MVN*). This approach assumes all imputed variables to follow a joint multivariate normal distribution. Conversely, MI by Chained Equations (MICE) is a semiparametric approach that does not take a joint MVN distribution but considers a different distribution for each imputed variable [62]. Unlike MVN, MICE employs a sequential (variable-by-variable) approach while incorporating functional relationships among variables and data characteristics such as ranges. Within MICE, one can either use Linear Regression or Predictive Mean Matching (PMM) for continuous variables. This article carries out the PMM technique to impute missing values. PMM relaxes most of the assumptions of parametric MI techniques

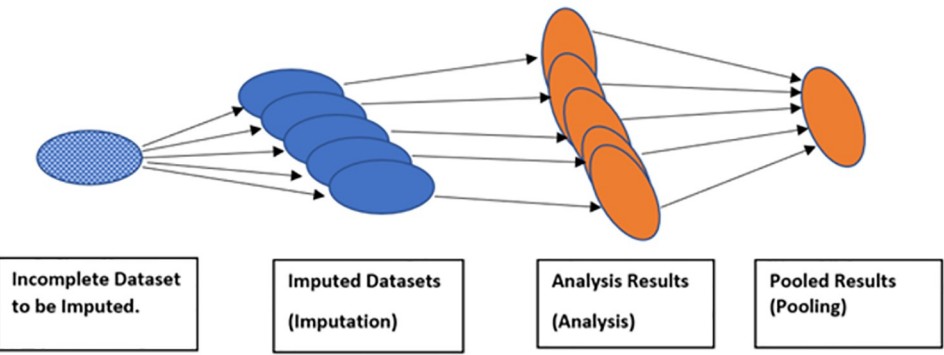

**Fig 1. Shows a standard multiple imputation process.** In the first step (imputation stage), missing data at hand, shown in white dots, are imputed (all in blue now showing imputation happened) to create *m* imputed datasets. Following imputation, each imputed dataset is separately analyzed using standard methods (such as OLS regression). Lastly, the analysis results are combined using Rubin's rules [22].

[30]. Hence, it is handy for imputing quantitative variables that are not normally distributed [63]. In the PMM, the missing value for an observation (considered as a 'recipient') is imputed by the observed value from another observation (called a 'donor') with a similar predicted mean outcome as follows [30, 64]:

In the *imputation* stage, for every missing value, the PMM algorithm structures a small set of donors (typically 5 or 10) from all complete cases that have *predicted* values closest to the predicted value for the missing value. Next, one donor is randomly drawn from the neighborhood pool. The observed value of such a donor is assigned to the missing value. This procedure is conducted *m* times, which generates *m* datasets. After the imputation stage, the *analysis* and *pooling* stages follow the same pattern as any standard MI. Like any MI, in the analysis stage *m* times analyses are conducted, and in the pooling stage, these results are combined to get a single estimate.

A more step-by-step computational process within the imputation stage of PMM is explained below:

Suppose there is a variable (X) that has missing values and another set of variables (Vs) to be used to impute X, the software (STATA or R) carries out the following computations in the imputation stage:

- Firstly, it estimates a linear regression of X on Vs for complete observations (those with no missing values). This step produces a set of coefficients *a*.

- Secondly, it randomly draws from the "posterior predictive distribution" of *a* (the posterior predictive distribution is the distribution of possible unobserved values conditional on the observed values [65]). This step generates a new set of coefficients $a^*$. (this step ensures variability in the imputed values produced later on).

- Thirdly, the software uses coefficients $a^*$ to generate predicted values for X for all observations.

- Fourthly, for each observation with a missing value of X, the software identifies a set of observations with observed X (called donors or neighbors) whose predicted values are roughly close or similar to the predicted value for the observation with missing data.

- Lastly, from the neighborhood pool identified, it randomly chooses one donor and designates its observed value to fill in for the missing value.

For each completed dataset, steps 2 through 5 are conducted. The key idea is constructing the right donor pool from where observations with missing data will be matched with observations with available data [66]. Researchers have answered how many donors or neighbors should be in the donor pool [29, 66]. They assert that the size of the pool depends on sample size. In general, for most situations, these studies suggest k = 10 or k = 5. The default in the Stata MI command is k = 1.

In short, PMM is simple to perform and a versatile method. It relaxes the normality distribution assumption, which is not always observed in continuous data. Since PMM imputations are based on observed neighborhood values, they are much more realistic. Unlike other techniques such as EM or MVN, PMM does not produce imputations outside the observed values; thus, they overcome the problems with meaningless imputations. Compared to other suites such as Normal Linear Regression imputation, PMM is also less susceptible to model specification and can handle many variables irrespective of their distributions [36]. While imputing from the neighboring donor candidates, it incorporates nonlinearities (nonlinear associations among variables) and returns the same distribution for missing data present in the observed data [36].

## 5. MSK panel dataset

Here I am presenting the main features of the MSK dataset. The dataset has been compiled and estimated after applying the MICE predictive mean matching technique described in the previous section. The complete dataset consists of information for many pertinent variables and for all LMICs eligible for IDA support over time (panel data). Specifically, the dataset contains complete data for 47 variables for 82 countries between 2005 and 2019 (1,230 country-year observations).

This new complete dataset offers ample statistical content to conduct longitudinal comparative country analyses of national absorptive capacity systems (NACS) within LMICs. Among other valuable insights, such analyses illustrate the relative standing of LMICs. Similarly, the dataset's time-series feature enlightens how LMICs' NACS evolved in the last one and a half decades. Immediate use of the dataset would entail estimating the relationship between the variables within the dataset (capacities constituting NACS) and the LMICs' economic development. Such an exercise will offer crucial lessons on economic growth and development to leading and lagging LMICs. Similarly, another use will involve clustering LMICs into different groups based on capacities scores.

Since NACS are multifaceted, any analysis of NACS would involve a large number of possibly relevant variables interacting in many ways. Therefore, the MSK dataset embraces a multi-dimensional operationalization of NACS. In this dataset, the NACS constitutes six capacities drawn from the literature. In addition, various *incoming* flows from abroad (learning, knowledge, skills, and technology) also may influence the NACS. Fig 2 represents these capacities of NACS while alluding to the incoming flows. The six capacities are: 1) Technological capacity, 2) Financial capacity, 3) Human capacity, 4) Infrastructural capacity, 5) Public Policy capacity, and 6) Social capacity. The discussion of all these capacities (and incoming flows) and how encompassing they are compared to other narrow definitions of capacities is beyond this article's scope (please see [12] for this discussion). However, the central hypothesized idea behind this dataset's construction is that LMICs that are severely lacking in data need to appreciate that these capacities and their dynamic interaction drive economic development and science, technology, and innovation (STI) in those economies. For this purpose, development

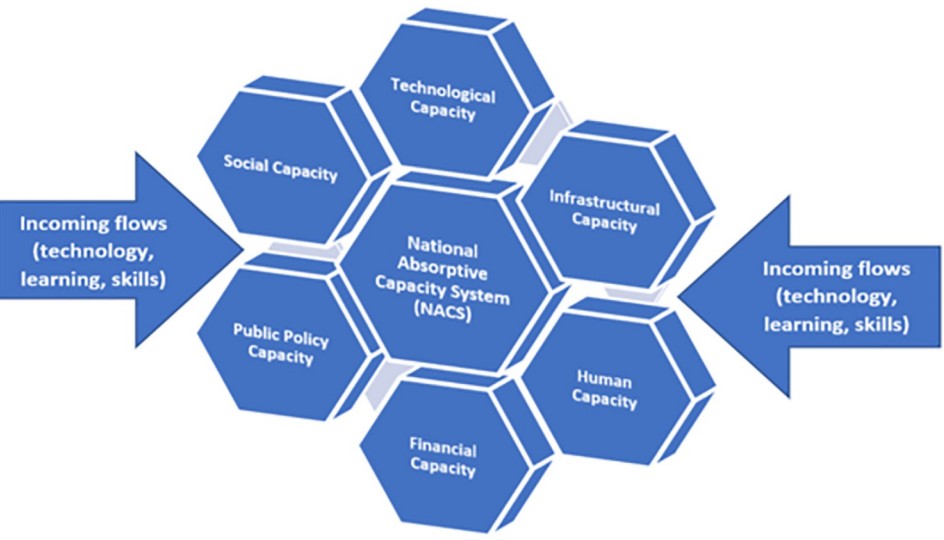

**Fig 2. Shows National Absorptive Capacity System (NACS) and its capacities.** These six capacities constitute NACS. Incoming flows mediate capacities within NACS. Figure Source: Khan [12].

economists and STI policymakers need access to panel statistical data (country-year observations) on these capacities, which would help them conduct empirical analyses.

Literature on NIS helped identify 64 variables, likely constituting one of these capacities in NACS. After performing imputation analysis, the list of variables was reduced. Resultantly, the MSK dataset consists of 47 variables, as shown in Table 1. As a matter of good practice, the table also compares descriptive statistics (mean, standard deviation, minimum, maximum, and observation count) of the variables in the new (complete) dataset with descriptive statistics for corresponding variables in the observed (incomplete) dataset. The last column of the table reports the share of missing data present in the original dataset. As can be seen, the missingness is very high for some variables; missingness ranges from 0.89% to about 87%. A quick look at the table shows that descriptive statistics of the two data (complete and incomplete) do not differ much. This is one of the many ways to show that the complete dataset is sufficiently reliable (this will be elaborated on in the forthcoming section).

The dataset was constructed in five main steps (also illustrated in S1 Fig).

## Step1- Data collection

In the first step, I collected 64 variables from publicly available databases (see S2 Table for a complete list of variables and their sources). These variables are potentially crucial for measuring the six capacities of countries. This initial dataset (original) contains a large number of missing values for countries and variables of interest.

## Step2- Choice of specification

To multiply impute, the choice of a correct multiple imputation specification is necessary. In *STATA*, either one can employ multivariate normal (MVN) MI or MI by chained equations (MICE). One can employ Amelia II in R statistical tool [54]. However, Amelia II assumes normality, which is not the case here. Both MVN and MICE strategies assume a MAR missing pattern in data before execution. There is no direct way of testing this (and other missingness MCAR and MNAR) assumption(s) illustrated earlier [67]. To repeat, MCAR is the most restrictive assumption (requiring the missingness to be truly random) whereas MNAR is the most relaxed (requiring the missingness to be systematic). Between these two assumptions is MAR, which is a moderately restricted assumption involving some degree of randomness.

Since it is hard to directly test for the assumptions, all one can do is offer a plausible explanation of how the missingness pattern may conform to MAR assumption or how it may not be MNAR and MCAR. Alternatively, one may investigate indirectly, including sensitivity analyses and multiple reliability checks of the imputed dataset, that are far from perfection [67]. One indirect way to check for MAR assumption is to implement some other imputation model that requires MNAR assumption and then compare the estimates (and standard errors) from both MAR and MNAR analyses (as conducted by [68]). The authors obtained approximately similar estimates in both analyses, thus claiming that their MAR assumption was adequate to get unbiased estimates [68]. However, it is hard to support this claim fully—the same estimates from different models may not mean unbiased estimates.

In addition to performing an imperfect sensitivity analysis of the dataset, another way would be to do reliability checks of the imputed dataset, just like the author has performed in this paper (the authors of the CANA dataset [31] also performed reliability checks). If the imputed dataset is reliable, it may imply that the MAR assumption was reasonably met and the bias in estimates is most likely mitigated. Another indirect way (ex-post) would be to compare the analysis results obtained through imputed (under MAR) and observed (incomplete) datasets, which may offer interesting retrospective insights about the quality of the dataset.

**Table 1. Descriptive statistics of new MSK dataset vs. incomplete observed dataset (for more details on the variables, please consult S2 Table).**

| Capacity and Variables | Variable code | MSK Dataset | | | | | Observed Dataset | | | | | Missing % |
|---|---|---|---|---|---|---|---|---|---|---|---|---|
| | | Obs | Mean | Std. Dev. | Min | Max | Obs | Mean | Std. Dev. | Min | Max | |
| **TECHNOLOGY CAPACITY** | | | | | | | | | | | | |
| Sci & tech. articles | tscitjar | 1230 | 1270.77 | 9395.79 | 0 | 135787.8 | 1,148 | 1236.60 | 9247.52 | 0 | 135787.8 | 6.67% |
| Intellectual payments (mil) | Tippay | 1230 | 65.35 | 492.20 | -13.92 | 7906 | 818 | 87.80 | 601 | -13.97 | 7909 | 33.50% |
| Voc. & tech. students (mil) | tsecedvoc | 1230 | 111698.6 | 253483.79 | 0 | 2300769 | 571 | 121436.2 | 277829.5 | 0 | 2300769 | 53.58% |
| R&D expend. % of GDP | Trandd | 1230 | .21 | .16 | .01 | .86 | 225 | 0.25 | 0.19 | 0.01 | 0.859 | 81.71% |
| R&D researchers (per mil) | Tresinrandd | 1230 | 162.65 | 225.9 | 5.94 | 1463.77 | 148 | 256 | 317 | 5.93 | 1463.77 | 87.97% |
| R&D technicians (per mil) | Ttechinrandd | 1230 | 57.02 | 63.01 | .13 | 627.73 | 144 | 55.27 | 70.22 | 0.13 | 627.73 | 88.29% |
| High-tech exports (mil) | Thigexperofmanex | 1230 | 6.23 | 9.29 | 0 | 68.14 | 547 | 5.80 | 8.74 | 0.00008 | 68.14 | 55.53% |
| ECI (econ. complexity) | teciscore | 1230 | -.72 | .63 | -3.04 | .82 | 892 | -0.77 | 0.62 | -3.04 | 0.82 | 27.48% |
| **FINANCIAL CAPACITY** | | | | | | | | | | | | |
| Tax revenue (% of GDP) | Ftaxrpergdp | 1230 | 16.22 | 11.71 | 0 | 149.28 | 583 | 15.7 | 11 | 0.0001 | 149.28 | 52.60% |
| Business startup cost | fcosbstpropergni | 1230 | 85.38 | 137.76 | 0 | 1314.6 | 1,154 | 79 | 120.2 | 0 | 1314.6 | 6.18% |
| Domestic credit by banks | Fdomcrprsecbybkpergdp | 1230 | 25.07 | 20.37 | .5 | 137.91 | 1,100 | 26.3 | 20.85 | 0.5 | 137.91 | 10.57% |
| Days to start business | ftdaystobusi | 1230 | 35.34 | 37.71 | 1 | 260.5 | 1,154 | 34.48 | 36.45 | 1 | 260.5 | 6.18% |
| Days enforcing contract | fdaystoenfctt | 1230 | 666.61 | 329.52 | 225 | 1800 | 1,154 | 662.2 | 322.4 | 225 | 1800 | 6.18% |
| Days to register property | Fdaystoregpro | 1230 | 87.33 | 97.58 | 1 | 690 | 1,104 | 81 | 89.6 | 1 | 690 | 10.24% |
| Openness measure | Fopenind | 1230 | .11 | .08 | .01 | .44 | 847 | 0.11 | 0.08 | 0.009 | 0.44 | 31.14% |
| Days to electric meter | Fdaystoobtelecconn | 1230 | 37.24 | 33.64 | 2.5 | 194.3 | 153 | 34.3 | 31.31 | 2.5 | 194.3 | 87.56% |
| Business density | fnewbusdenper1k | 1230 | 1.06 | 1.47 | .01 | 12.31 | 583 | 1.19 | 1.67 | 0.006 | 12.30 | 52.60% |
| Financial accountholders | faccownperofpop15p | 1230 | 30.94 | 22.53 | 1.52 | 92.97 | 160 | 30 | 19.28 | 1.52 | 92.97 | 86.99% |
| Commercial banks | fcombkbr1k | 1230 | 10.49 | 11.99 | .27 | 71.23 | 1,099 | 10.58 | 12.045 | 0.27 | 71.23 | 10.65% |
| **HUMAN CAPITAL CAPACITY** | | | | | | | | | | | | |
| Primary enrollment (gross) | hprimenrollpergross | 1230 | 103.36 | 18.18 | 23.36 | 149.96 | 911 | 103.4 | 18.15 | 23.36 | 149.95 | 25.93% |
| Sec. enrollment (gross) | hsecenrollpergross | 1230 | 57.49 | 25.99 | 5.93 | 123.03 | 711 | 58.03 | 26.63 | 5.93 | 123.03 | 42.20% |
| Primary pupil-teacher ratio | Hpupteapriratio | 1230 | 34.43 | 14.36 | 8.68 | 100.24 | 751 | 35.3 | 14.63 | 8.68 | 100.24 | 38.94% |
| Primary completion rate | Hprimcompra | 1230 | 79.41 | 20.89 | 26.1 | 134.54 | 735 | 78.83 | 20.72 | 26.09 | 134.54 | 40.24% |
| Govt. expend. on educ. | Hgvtexpedupergdp | 1230 | 4.36 | 2.22 | .69 | 12.9 | 615 | 4.06 | 1.91 | 0.69 | 12.90 | 50% |
| Human Capital Index 0–1 | hhciscale0to1 | 1230 | .42 | .09 | .29 | .69 | 154 | 0.43 | 0.09 | 0.28 | 0.69 | 87.42% |
| Advanced educ. labor | Hlfwithadedu | 1230 | 75.5 | 10.55 | 39.97 | 96.36 | 265 | 76.08 | 10.29 | 40 | 96.36 | 78.46% |

*(Continued)*

**Table 1.** (Continued)

| Capacity and Variables | Variable code | MSK Dataset | | | | | Observed Dataset | | | | | Missing % |
|---|---|---|---|---|---|---|---|---|---|---|---|---|
| | | Obs | Mean | Std. Dev. | Min | Max | Obs | Mean | Std. Dev. | Min | Max | |
| Compulsory educ. (years) | hcompeduyears | 1230 | 8.45 | 2.16 | 4 | 15 | 1,028 | 8.57 | 2.16 | 4 | 15 | 16.42% |
| Industry employment | Hempinduspertotem | 1230 | 14.52 | 7 | .64 | 32.59 | 1,125 | 14.08 | 6.94 | 0.64 | 32.59 | 8.54% |
| Service employment | Hempserpertotem | 1230 | 39.43 | 15.05 | 7.16 | 75.34 | 1,125 | 37.8 | 14.24 | 7.16 | 75.34 | 8.54% |
| **INFRASTRUCTURE CAPACITY** | | | | | | | | | | | | |
| Mobile subscriptions | imobsubper100 | 1230 | 59.12 | 38.15 | .26 | 181.33 | 1,219 | 59.19 | 38.17 | 0.26 | 181.33 | 0.89% |
| Access to electricity | Iacceselecperpop | 1230 | 57.02 | 31.3 | 1.24 | 100 | 1,135 | 56.77 | 31.32 | 1.24 | 100 | 7.72% |
| Broadband subscriptions | ibdbandsubper100 | 1230 | 1.97 | 4.12 | 0 | 25.41 | 1,114 | 2.02 | 4.23 | 0 | 25.41 | 9.43% |
| Telephone subscriptions | itelesubper100 | 1230 | 5.31 | 7.39 | 0 | 32.85 | 1,218 | 5.29 | 7.40 | 0 | 32.85 | 0.98% |
| Energy use (per capita) | Ienergyusepercap | 1230 | 560.21 | 392.9 | 9.55 | 2246.92 | 471 | 553 | 376.25 | 9.54 | 2246.92 | 61.71% |
| Logistic perf. Index 1–5 | ilpiquoftratraninfr | 1230 | 2.18 | .33 | 1.1 | 3.34 | 372 | 2.19 | 0.32 | 1.1 | 3.34 | 69.76% |
| Internet users | Iindintperpop | 1230 | 16 | 16.3 | .03 | 89.44 | 1,209 | 16 | 16.33 | 0.031 | 89.44 | 1.71% |
| **PUBLIC POLICY CAPACITY** | | | | | | | | | | | | |
| CPIA econ. mgmt. | pcpiaeconmgtcl1to6 | 1230 | 3.39 | .69 | 1 | 5.5 | 1,132 | 3.40 | 0.67 | 1 | 5.5 | 7.97% |
| Public sect. mgmt. & instit | pcpiapsmgandinscl1to6 | 1230 | 3.06 | .5 | 1.4 | 4.2 | 1,132 | 3.06 | 0.48 | 1.4 | 4.2 | 7.97% |
| Sructural policies | pcpiastpolclavg1to6 | 1230 | 3.3 | .54 | 1.17 | 5 | 1,132 | 3.31 | 0.52 | 1.17 | 5 | 7.97% |
| Statistical capacity 0–100 | Pscapscoravg | 1230 | 59.82 | 14.89 | 20 | 96.67 | 1,206 | 59.9 | 14.87 | 20 | 96.67 | 1.95% |
| Legal Rights Index 0–12 | pstrengthoflegalright | 1230 | 4.83 | 3.1 | 0 | 11 | 565 | 5.27 | 3.05 | 0 | 11 | 54.07% |
| **SOCIAL CAPACITY** | | | | | | | | | | | | |
| Human resources rating | scpiabdhumanres1to6 | 1230 | 3.52 | .63 | 1 | 4.5 | 1,132 | 3.52 | 0.61 | 1 | 4.5 | 7.97% |
| Equity of public resc use | scpiaeqofpbresuse1to6 | 1230 | 3.38 | .64 | 1 | 4.5 | 1,132 | 3.39 | 0.62 | 1 | 4.5 | 7.97% |
| Social protection rating | scpiasocprorat1to6 | 1230 | 3.03 | .59 | 1 | 4.5 | 1,128 | 3.04 | 0.58 | 1 | 4.5 | 8.29% |
| Social inclusion o. | scpiapolsocinclcl1to6 | 1230 | 3.28 | .51 | 1.5 | 4.3 | 1,129 | 3.28 | 0.50 | 1.5 | 4.3 | 8.29% |
| National headcount poverty | spovheadcnational | 1230 | 38.52 | 15.13 | 4.1 | 82.3 | 234 | 35.90 | 14.20 | 4.1 | 82.3 | 80.98% |
| Social contributions | Ssocialconperofrev | 1230 | 3.23 | 7.53 | 0 | 39.74 | 569 | 3.90 | 8.77 | 0 | 39.74 | 53.74% |

Here I argue missingness in LMICs exhibit a MAR pattern. In other words, the missingness pattern in data in LMICs is "somewhat" random (as opposed to "completely random" required by MCAR or "truly systematic" as can be seen in the MNAR pattern). The MAR pattern, by definition, implies that the observed data (in LMICs) can explain and predict missingness [59]. Therefore, to say that a dataset for LMICs has a MAR pattern, a researcher needs access to some available data in these countries.

To reiterate, I argue that LMICs have *most likely* MAR data pattern that is somewhat random but not entirely systematic. If the missing would have been all systematic (MNAR pattern), then we would not have rich data in such countries on other important economic,

geographic, seasonal, and demographic data. Since we have plenty of data for the parameters mentioned earlier in LMICs, the data are not MNAR. On the other hand, if missingness had been completely random (MCAR), i.e., neither missing data nor available data impact missingness in LMICs, the missing pattern in variables would have been uncorrelated. As the missingness pattern in LMICs is correlated in many or some variables, the data are not MCAR. Thus, the data in LMICs are MAR. LMICs can have missing data for various reasons, ranging from poor data infrastructures and meager resources to frequent natural disasters and severe civil conflicts. However, despite missingness in many variables of significance, the availability of rich information on poverty indicators, economic development, literacy rates, and demographics in LMICs can be useful. The propensity of missing values for essential variables (such as science and tech articles, budget allocation to education, and service sector employment, among other variables in the dataset) are systematically linked with the LMICs' observed data (*not the missing data*) on GDP, per capita income, and literacy rates variables. For instance, a country's per capita income and literacy rate relate to a country's allocation to education expenditure. Thus, I argue that the rich corpus of observed data can be employed to explain and predict the missingness pattern for data on other variables, as required by the MAR assumption.

Furthermore, since all the variables are continuous, differently distributed, and missingness among them is "somewhat" arbitrary, Rubin's [23] multiple imputation by chained equations (MICE) best serves this study. Researchers argue that MICE allows sound modeling for missing values and provides rigorous standard errors for the fitted parameters [62, 69]. MICE treats each variable with missing values as the dependent variable in a regression, with the remaining variables as its predictors. Once MICE is specified, as mentioned earlier, within MICE, one can use either a linear regression *(regress)* or predictive mean matching (*PMM*) specification for continuous variables. Chained imputation with *linear regression* has a severe pitfall as it implements normal distribution on imputed values regardless of the distribution of original values [69]. Conversely, PMM caters to this problem by respecting the observed values' distribution pattern. Besides, the use of PMM is robust against other misspecifications in the imputation model [63]. Notably, it is robust against heteroskedastic residuals and nonlinear associations between variables [36, 63]. Since the observed variables are not normally distributed (see kernel density graphs plotted after imputation in Supporting information) and their residuals are heteroscedastic, PMM is the most suitable chained imputation for this data.

## Step3- Variable shortlisting and running the first round of imputations

In the third step, I ran MICE in *STATA 16* for all variables. Out of 64 variables, chained imputations did not work for three variables (multipoverty index, multipoverty intensity, agricultural machinery). The system gave the error message that "the posterior distribution from which MI drew the imputations for these variables is not proper when the VCE estimated from the observed data is not positive definite." This essentially means that there is collinearity. Since these variables have more than 97% missing values, I dropped off these variables from the analysis to deal with the reported error. I tried linear regression specification too, but again it did not work. Then I run a first successful round of imputations (m = 20) followed by descriptive analyses of all these 61 variables. Out of these variables, I dropped off another 14 variables because the results were not of sufficient reliability. They had a considerable fraction of missing information (FMI). Generally, these variables reported FMI higher than 60%. It is important to understand that FMI is the proportion of the total sampling variance that is due to missing data, and it is calculated based on the percentage missing for a specific variable and how correlated this variable is with other variables in the imputation model [70]. Besides a

higher FMI, the variables that had to be dropped off had their descriptive statistics very different from the observed (incomplete) dataset and varied greatly in successful imputations. Thus, overall, the list of variables was reduced to 47.

### Step 4- Running the second round of imputations on shortlisted variables

In the fourth step, I did a second round of PMM imputations for the truncated list of 47 variables together. I included data on complete variables of time and country identifiers (year and country) and auxiliary variables (GDP per capita, technical cooperation grant, total population, gross capital formation, net ODA and official aid assistance, number of international tourist arrivals receipts, merchandise import from high-income economies as percentage of total merchandize imports, current health expenditure) following the recommendations of the multiple imputation literature. The inclusion of complete identifiers and other auxiliary variables increases the precision of the imputation results for variables exhibiting high missingness and makes the MAR assumption more plausible statistically [71]. To obtain a high-efficiency level in parameter results, I set m = 50, i.e., fifty complete datasets (copies of the original dataset) were estimated for all 47 variables.

While traditionally researchers set m = 5 or 10, new research indicates that m should be high to achieve accurate standard errors and point estimates [72]. With large m, variance estimates stabilize, and standard errors become more accurate. In essence, by returning accurate standard errors, large m models the uncertainty within imputations (missing values are uncertain) with more certainty. In addition, large m is particularly recommended if FMI is high for variables. Similarly, large m increases the relative efficiency of parameters (point estimates). i.e., how well the true population parameters are estimated. Generally, when the amount of missing information is high, more imputations (high m) are needed to attain adequate efficiency for point estimates [70, 72].

After setting m, subsequent econometric analyses are performed separately on each dataset (50 analyses because m = 50). Then, the results from each analysis are pooled according to Rubin's rules. Here, I randomly pick results from imputation # 25 for descriptive statistics and illustration purposes. This dataset contains 47 variables for 1,230 observations (82 countries for the period 2005–2019).

### Step 5- Quality check

Finally, I thoroughly investigated the variables to analyze the imputed values' quality. This investigation informs the extent to which the new complete dataset may be regarded as reliable. I did a visual inspection of kernel density graphs of imputed, completed, and original values for all the variables in this investigation. Similarly, I checked descriptive statistics of observed and imputed values. This quality check is discussed fully in the next section. This check results suggest that multiple imputations with PMM have been successful for the truncated list of variables.

In brief, following the above steps, the final version of the MSK database is constructed and made available. The dataset consists of 47 variables for 82 IDA-eligible countries spanning over 15 years (1,230 country-years observations). In contrast, the remaining 17 variables were rejected and not included in the database because either the system could not impute them or returned unreliable imputed values of poor quality.

## 6. Descriptive analysis of the MSK dataset

To empirically illustrate the usefulness of the MSK dataset and how it can be used to study absorptive capacity systems across countries, I have conducted a detailed analysis in another

article [12]. A brief descriptive analysis of the MSK dataset is conducted here. This analysis offers insights into the trends in capacities constituting NACS in LMICs and how they evolve over time. Three brief analyses are conducted: distribution (kernel density) of select few variables of interest within each capacity at the start, middle, and the end of the study period (i.e., 2005, 2010, and 2019); time trends (2005–2019) of the variables of interest for select countries (six countries, one from each region in our countries of study); and comparative ranking of countries based on composite capacity indices.

### i) Distribution (kernel density) of select few variables of interest within each capacity at different periods (i.e., 2005, 2010, and 2019)

The distribution patterns (S2 Fig) are drawn for a select set of variables from each capacity for three years (2005, 2010, and 2019). Distributions for technological capacity by and large show that LMICs have not significantly improved their technological base. A rightward shift in distributions for infrastructure capacity indicates that LMICs overall have experienced an improvement in their infrastructure base. However, we see a leftward shift in the distributions for social capacity, meaning that LMICs eligible for IDA support are moving backward in their social capacity. For the remaining three capacities (human, financial, and public policy), cross-country distributions' evolution is not very evident. Their pattern depends on the specific variable under discussion. For example, distributions for human capacity show that employment in the service sector has improved over time. On the contrary, expenditure on education has not increased.

### ii) Time trends (2005–2019) of the variables of interest for select countries (six countries, one from each region in our countries of study)

Next, time trends of the select variables from each capacity are observed over time for six countries (S3 Fig). The trends for technological capacity variables vary over time for most countries. In Pakistan, while most trends in such variables are either uniform or erratic, the trends in scientific articles and ECI scores rise. Similar trends (uniform in some cases and unpredictable in others) are observed for financial capacity variables. Myanmar and Nicaragua experience a rising trend in domestic credit availability, while other countries have experienced an oscillating trend (increasing and then decreasing). In the case of human capacity and infrastructure capacity, trends for some variables (primary completion, expenditure on education, LPI score) have experienced erratic movements; however, most countries are improving in other variables (service and technological sector employment, mobile and internet penetration) of these capacities. This may allude to the fact that these countries are perhaps catching up with advanced economies in terms of these indicators. Finally, it is hard to identify a clear winner for the last two capacities (public policy and social capacity); most trends are either uniform or erratic. However, the statistical score index is strikingly improving for Djibouti and Myanmar. These results largely corroborate the abovementioned distribution analysis. The crux is that countries show varying progress (clearly visible in some cases and diffused in others) over time for all these variables.

### iii) Comparative ranking of countries

Lastly, a comparative ranking of countries was conducted for recent data in 2019 (see S3 Table). For this, I first calculated six composite indices (Technology, Finance, Human Capital, Infrastructure, Public Policy, and Social Capacity) and then aggregated them into a composite Absorptive Capacity Index. Vietnam tops the list of the countries, whereas South Sudan scored

the least. This ranking can be conducted for all years, which would show longitudinal changes in absorptive capacity systems of countries.

While not an exhaustive list of the uses of the dataset, these analyses provided a flavor of how this dataset might be used in comparative analyses of National Absorptive Capacity Systems. These analyses can be extended and conducted in several ways in future research. This section's purpose was to demonstrate how one might get started on subsequent empirical analyses.

## 7. Quality check of the estimated MSK dataset

A quality check is conducted to determine the usefulness and vitality of this dataset.

As mentioned in section 5, I collected 64 variables to measure countries' capacities to construct the database. After carrying out imputations and evaluation, I shortlisted 47 variables to be included in the dataset for an entire range of 1,230 country-year observations (15 years for 82 countries). The remaining 17 variables were rejected either because the system could not impute them (three variables) or the results produced (14 variables) were not of good quality.

In order to assess the imputation procedure and the reliability of the variables included in the MSK dataset, this article conducts a four-way quality check: first descriptive statistics of the two datasets (complete and observed) are conducted; secondly, distributions of completed and observed datasets are observed; thirdly, correlation tables of the observed and complete variables are compared; and fourthly, trends within imputations and convergence pattern are observed.

### i) Descriptive statistics of two datasets

I looked into means, maximum, minimum, and standard deviation for complete and observed datasets. Table 1 reports a comparison of such descriptive statistics for both datasets. First, the table indicates that means (averages) and standard deviations (variability) for all 47 variables are almost identical. Imputing at the mean might reduce variability in some variables, though (as evident in lower standard deviation values). Secondly, we can see that the complete dataset has the same maxima and minima, and the values are meaningful (no negative numbers on researchers, for instance). Moreover, I inspected relative efficiency values for only imputed variables. This glance of relative efficiency values (above 98% for all variables with m = 50) suggested highly efficient point estimates. All this shows that the complete dataset's imputed values are roughly the best approximation of the original sources' missing data.

### ii) Distribution of compete and observed dataset

A detailed distribution assessment is conducted for the two datasets. This is accomplished via visual inspection of kernel densities for all 47 variables in the observed (incomplete) and complete (MSK) datasets.

The logic behind comparing the two datasets' statistical distributions is to see how best the complete dataset is an extension of the observed dataset. If the two distributions are roughly similar, we can claim the reliability of the imputed values. But, if the two distributions differ, the imputation results may not be reliable.

Visual inspection of kernel densities provides an interesting quality check (See S4 Fig). I looked into kernel density distributions at different imputations (randomly chosen) for all capacities. For almost all the variables within capacities, variables' distributions in the MSK dataset are similar to those in the incomplete data in various imputations. Even for those variables that report missingness higher than 80% (R & D, Researchers, Technicians, Account ownership, HCI scale), the approximation level (similarity), while relatively lower, is still very

close to the original distributions. This means that the PMM imputation has successfully estimated missing information with high accuracy. Thus, this visual inspection of kernel density distributions grants substantial reliability status to the MSK dataset.

### iii) Correlation tables of the original and complete

Lastly, pairwise correlation coefficients are calculated and compared in the original dataset (m = 0) and complete dataset (at imputation m = 25). The S4 Table(s) report such correlation coefficients for each capacity within both datasets. The correlation coefficients for the observed dataset are reported above the pairwise correlations for the complete dataset.

The rationale behind this correlation comparison is if the two correlations are similar, then statistical distributions between the two will likely match. This will indicate the reliability of the imputation results. However, if the two coefficients are not comparable, this would mean unreliability and bias in the imputation results produced through the imputation procedure. The bias and unreliability will subsequently affect the post-imputation analysis on the complete dataset.

A close inspection of the correlation tables suggests that correlation coefficients are very similar across the variables in both datasets. Not only the magnitudes of coefficients are roughly similar, but also the signs of the coefficients are maintained in the complete dataset following the multiple imputation exercise. Some coefficients (for example, R & D, Number of technicians, and Domestic credit, among others) change in size; however, these changes are not substantial. Overall, this check suggests that PMM imputation has preserved the correlation structure among the variables. Thus, it can be concluded that the MSK dataset is sufficiently reliable.

### iv) Trends within imputations and convergence pattern

Similarly, I inspected the trends in imputed variables' values across imputations (at m = 1, m = 10, m = 25, m = 40, m = 50). I noticed that values across imputations were highly similar, suggesting that the imputation exercise was successful. Also, since the dataset was obtained through chained imputations involving iterations, the reliability of the imputation process must be established. Therefore, to establish the reliability of the imputation process, I checked for convergence among iterations for imputed variables. Convergence can be checked in a few ways. One way is to plot the mean and variance of the imputed values of different missing variables against the iteration number [73]. For healthy convergence, these plots for *m* imputed datasets should freely intermingle, and there should not be any definite trends [73, 74]. Another way is to examine between and within sequence variance [73, 74]. On healthy convergence, the variance between sequences is no larger than the variance within each sequence [73–75]. Since the plots for imputed datasets freely intermingled with no definite trend, the convergence pattern of the iterations through which the dataset was generated showed a healthy convergence (S5 Fig). All this shows that the MSK dataset is of good quality.

## 8. Conclusion and implications

Comparative country analyses on absorptive capacity and economic development in LMICs lack because of the lack of complete data availability. To address this problem, this article employed Rubin's Multiple Imputation to impute missing values in variables. Specifically, it used Multiple Imputation by Chained Equations with a Predictive Mean Matching approach to estimate the MSK panel dataset. The dataset consisted of six country-related capacities. A total of 47 continuous variables measured these capacities. This dataset was estimated from an

observed dataset containing many missing values. The complete dataset contained 82 countries for the period 2005–2019, for 1,230 country-year observations.

The MSK dataset provides a rich panel (across countries and over time) of statistical content that can be used in several ways. For instance, this dataset can be used to estimate the impact of absorptive capacities on economic growth in LMICs. Similarly, the capacities can be aggregated for different LMICs to find the relative standing of one economy vis-à-vis other economies. Further, such an exercise can be used to investigate the factors of development within leading and lagging LMICs. Finding leading and lagging economies within LMICs at the same level of development offer lessons to lagging economies on how they can catch up. Here, I demonstrated how a simple descriptive analysis of capacities within the complete dataset could be used to gain insights into the dynamic evolution of such capacities in different countries.

On the methodological front, MICE PMM for estimating dataset for the comparative analyses of capacities and economic growth in LMICs is powerful compared to other solutions such as mean imputation or deletion. MICE PMM is powerful because it retains variability in data as the imputed value is randomly taken from the suitable donor pool. Moreover, PMM is a good technique because it reduces bias by keeping information on all variables: variables for which partial data is available are imputed rather than deleted. Similarly, the technique preserves representation (by keeping all economies even if they have partial data rather than dropping them of analysis), returns accurate or realistic data (imputed data is taken from neighboring data pool), and captures dynamic evolution for all economies (which is compromised by using other imputation techniques).

However, MI returns multiple datasets, which indicates the uncertainty underlying missing data values. Thus, no matter how rigorous MI is, no imputation can claim with 100 percent certainty the accuracy of imputed values. Therefore, the dataset generated through MI must be carefully used for any analysis. The results of such analysis must make a disclaimer about the process through which the dataset was obtained. The reliability or quality check must be performed on the newly generated dataset, just as conducted for the MSK dataset. The MSK dataset generated here passed the quality check as the observed and complete dataset exhibited almost similar distributions, descriptive statistics, and correlation coefficients, and the process through which the dataset was imputed returned a healthy convergence among iterations.

As the MI-generated dataset is reliable, such a dataset can be valuable for hypothesis generation in LMICs suffering from poor data environments. Results from analyses based on original datasets for countries (and LMICs) with reasonably complete datasets can be compared with those based on imputed datasets. This may give some insights into the relative vitality of the completed dataset alongside interesting findings on what drives economic development in various countries. Moreover, future research may estimate datasets generated assuming MNAR pattern and then compare the datasets from both MAR and MNAR analyses to further investigate the strength of the MSK dataset.

## Supporting information

**S1 Table. Handling missing data strategies, assumptions, advantages and disadvantages.**
(DOCX)

**S2 Table. List of all 64 variables, their definitions, sources, missingness amount in observed variables, and acceptance/rejection status for the MSK dataset.**
(DOCX)

**S3 Table. Comparative ranking of countries as per absorptive capacity index (2019).**
(DOCX)

**S4 Table. Pairwise correlations for incomplete (m = 0) and complete datasets (m = 25).**
(DOCX)

**S1 Fig. Construction of the MSK dataset.**
(DOCX)

**S2 Fig. Kernel densities for select variables of interest at different points.**
(DOCX)

**S3 Fig. Time trends for select countries for select variables.**
(DOCX)

**S4 Fig. Kernel densities of the observed and complete dataset.**
(DOCX)

**S5 Fig. Checking for convergence through trace plots.**
(DOCX)

## Acknowledgments

This article was presented at the 17th Globelics International Conference, Heredia, Costa Rica, November 2021, the 35th Annual General Meeting and Conference, Pakistan Society of Development Economics, Pakistan Institute of Development Economics in Peshawar, Pakistan, November 2021, Shanghai University of International Business and Economics, Institute of Artificial Intelligence and Change Management, in Shanghai, China (online), December 2021, and the Schar School of Policy and Government, George Mason University, in Virginia, US, June 2022. I am highly grateful to the conference participants, the anonymous referees, and the editor of this journal for their valuable comments and suggestions. I also want to thank David M. Hart, James L. Olds, Maurice D. Kugler, and Lucas Núñez for their feedback on this article.

## Author Contributions

**Conceptualization:** Muhammad Salar Khan.

**Data curation:** Muhammad Salar Khan.

**Formal analysis:** Muhammad Salar Khan.

**Funding acquisition:** Muhammad Salar Khan.

**Investigation:** Muhammad Salar Khan.

**Methodology:** Muhammad Salar Khan.

**Project administration:** Muhammad Salar Khan.

**Resources:** Muhammad Salar Khan.

**Software:** Muhammad Salar Khan.

**Supervision:** Muhammad Salar Khan.

**Validation:** Muhammad Salar Khan.

**Visualization:** Muhammad Salar Khan.

**Writing – original draft:** Muhammad Salar Khan.

**Writing – review & editing:** Muhammad Salar Khan.

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
