## [Decision Letter · Decision Letter 0]

17 Jun 2022

PONE-D-21-35486Estimating a new panel MSK dataset for comparative analyses of national absorptive capacity systems, economic growth, and development in low and middle income economiesPLOS ONE

Dear Dr. Khan,

Thank you for submitting your manuscript to PLOS ONE. After careful consideration, we feel that it has merit but does not fully meet PLOS ONE’s publication criteria as it currently stands. Therefore, we invite you to submit a revised version of the manuscript that addresses the points raised during the review process.

We look forward to receiving your revised manuscript.

Kind regards,

Qaiser Abbas

Academic Editor

PLOS ONE

**Journal requirements:**

2. We noted in your submission details that a portion of your manuscript may have been presented or published elsewhere. [DETAILS AS NEEDED] Please clarify whether this [conference proceeding or publication] was peer-reviewed and formally published. If this work was previously peer-reviewed and published, in the cover letter please provide the reason that this work does not constitute dual publication and should be included in the current manuscript.

Reviewers' comments:

Reviewer's Responses to Questions

**Comments to the Author**

1. Is the manuscript technically sound, and do the data support the conclusions?

Reviewer #1: Yes

Reviewer #2: Yes

2. Has the statistical analysis been performed appropriately and rigorously? 

Reviewer #1: I Don't Know

Reviewer #2: Yes

3. Have the authors made all data underlying the findings in their manuscript fully available?

Reviewer #1: Yes

Reviewer #2: Yes

4. Is the manuscript presented in an intelligible fashion and written in standard English?

Reviewer #1: Yes

Reviewer #2: Yes

5. Review Comments to the Author

Reviewer #1: Just to clarify on the first footnote…when you are talking about relative poverty, the first dollar threshold that you mention is the world poverty threshold ($1,185 for 2021). Is that correct? And I am also assuming that is U.S. dollars.

Just a small issue, but on page 3 you mention NIS and NACS as being “fields”. My guess is that they should more correctly be referred to as organizations.

On page 3, you mention “state-of-the-art statistical methods to address the missing data problem”. You can probably elaborate a little further here as to what these authors used. I am guessing that some may be different from the predictive mean matching (PMM) technique that you use.

Your first reference to the MSK data set occurs on page 4 of the document. I believe that this is the proposed data set which you are building. Perhaps you can introduce it by calling it the “proposed” MSK data set.

Check out the second paragraph, sentence 3 on page 4. You actually refer to it as M1 rather than MI.

On page 4, you mention that “Public Policy and Social Capacity are operationalized very differently”. The reader may be curious as to how that is. You might want to consider elaborating on that.

On page 5, you mention that “The incomplete (original or observed) dataset is constructed from reputable data sources and contains many missing values.” Some elaboration on these data sources might be necessary here.

You mention the word “missingness” throughout the document. I am assuming that is a commonly used term in the imputation world.

At the bottom of page 8, replace “resulting results” with just “results”.

I appreciate the thoroughness with which you have constructed the data set. Still, I harken back to my days within a data science classroom where my professor stated that the rule of thumb for whether or not to provide imputations within a model was 10%...in other words, if a variable had more than 10% of its data missing, then you should just drop it from a model altogether. Here, the stakes seem even higher, in that you are building theoretical database for even more researchers to use in the future, based on your imputation strategy. Clearly, from Table 1, you can see that quite a few variables have far more than 10% of the data missing. I do appreciate your efforts in choosing the right imputation strategy here, and I do not take issue with the steps you have taken. However, I would be much more relieved if you could point to similar instances in which researchers have done a complex imputation strategy such as yours, and produced a viable data set that has been replicable and well-utilized by the research community in the past. Since Rubin (1987) is the seminal work on this issue, and you cite it liberally throughout the manuscript, I am tempted to agree with your work here but having precedent would certainly be nice to see.

On page 19, you make the following statement: “I have conducted a detailed analysis in another article (Khan 2021).” Just a small issue in that appears to be a working paper. It is nitpicky, but it has subtle differences.

Reviewer #2: Referee report: Estimating a new panel MSK dataset for comparative analyses of national absorptive capacity systems, economic growth, and development in low and middle income economies

Summary: This paper proposes a new complete panel dataset with no missing values for the low- and middle-income countries eligible for World Bank’s International Development Association’s support, by using Predictive Mean Matching developed by Rubin (1987). I think that this paper addresses an important issue (data limitation) and contributes to the literature by providing new data set. I think that the author needs to support the pivotal assumptions of the method he or she used.

1. The Multiple Imputation Method requires MAR (Missing at Random) assumption in page 10, according to the author. However, the author didn’t justify this assumption enough in the draft. The author just says “I argue that this rich corpus of data can be employed to explain and predict the missingness pattern for data on other variables, thus justifying the MAR assumption” and “The inclusion of complete identifiers and other auxiliary variables increases the precision of the imputation results for variables exhibiting high missingness and makes the MAR assumption more plausible”. The author also says “Missing At Random (MAR)- Data exhibits MAR if the missingness is due to observed but not unobserved data. In other words, the observed data explains the missingness.”. I don’t understand how merely having rich data set can support the argument that no unobserved data causes the missingness.

a. It is understandable that it is impossible to test MAR assumption directly. However, the author may be able to find other justification or indirect test to check this assumption.

2. In page 17, the author says “Furthermore, since all the variables are continuous, differently distributed, and missingness among them is arbitrary, Rubin’s (1987) multiple imputation by chained equations (MICE) best serves this study.”. It may be true that all variables are continuous, differently distributed. However, I don’t see the reason why missingness among them is “arbitrary”.

a. The author may define the meaning of “arbitrary” in this case and why missingness is “arbitrary” in this case.

b. If “arbitrary” means MAR assumption, then the aforementioned sentence (all the variables are continuous, differently distributed, and missingness among them is arbitrary, Rubin’s (1987)….) cannot be meaningful since the author didn’t justify the MAR assumption enough.

3. In page 23, it says “a health convergence” and “A healthy convergence means that variance between and within iterations is the same”. The author needs a citation.

6. PLOS authors have the option to publish the peer review history of their article (what does this mean?). If published, this will include your full peer review and any attached files.

Reviewer #1: No

Reviewer #2: No

---

## [Decision Letter · Decision Letter 1]

28 Aug 2022

Estimating a panel MSK dataset for comparative analyses of national absorptive capacity systems, economic growth, and development in low and middle income countries

PONE-D-21-35486R1

Dear Dr. Muhammad Salar Khan,

We’re pleased to inform you that your manuscript has been judged scientifically suitable for publication and will be formally accepted for publication once it meets all outstanding technical requirements.

Kind regards,

Larissa-Margareta Batrancea

Academic Editor

PLOS ONE

Additional Editor Comments (optional):

Reviewers' comments:

Reviewer's Responses to Questions

**Comments to the Author**

1. If the authors have adequately addressed your comments raised in a previous round of review and you feel that this manuscript is now acceptable for publication, you may indicate that here to bypass the “Comments to the Author” section, enter your conflict of interest statement in the “Confidential to Editor” section, and submit your "Accept" recommendation.

Reviewer #1: All comments have been addressed

Reviewer #2: All comments have been addressed

2. Is the manuscript technically sound, and do the data support the conclusions?

Reviewer #1: Yes

Reviewer #2: Yes

3. Has the statistical analysis been performed appropriately and rigorously? 

Reviewer #1: Yes

Reviewer #2: Yes

4. Have the authors made all data underlying the findings in their manuscript fully available?

Reviewer #1: Yes

Reviewer #2: Yes

5. Is the manuscript presented in an intelligible fashion and written in standard English?

Reviewer #1: Yes

Reviewer #2: Yes

6. Review Comments to the Author

Reviewer #1: I appreciated that the authors took the time to answer my inquiries so thoroughly. This manuscript seems ready to go for publication.

Reviewer #2: I think that the authors addressed all issues I raised. Even though your revision for one issue (testing assumption of MAR) is not perfect, but this is because the assumptions cannot be tested directly.

7. PLOS authors have the option to publish the peer review history of their article (what does this mean?). If published, this will include your full peer review and any attached files.

Reviewer #1: No

Reviewer #2: No

---

## [Editor Report · Acceptance letter]

12 Oct 2022

PONE-D-21-35486R1 

Estimating a panel MSK dataset for comparative analyses of national absorptive capacity systems, economic growth, and development in low and middle income countries 

Dear Dr. Khan:

I'm pleased to inform you that your manuscript has been deemed suitable for publication in PLOS ONE. Congratulations! Your manuscript is now with our production department. 

Kind regards, 

on behalf of

Dr. Larissa-Margareta Batrancea 

Academic Editor

PLOS ONE